# Successful Bacteriophage-Antibiotic Combination Therapy against Multidrug-Resistant *Pseudomonas aeruginosa* Left Ventricular Assist Device Driveline Infection

**DOI:** 10.3390/v15051210

**Published:** 2023-05-20

**Authors:** Karlis Racenis, Janis Lacis, Dace Rezevska, Laima Mukane, Aija Vilde, Ints Putnins, Sarah Djebara, Maya Merabishvili, Jean-Paul Pirnay, Marika Kalnina, Aivars Petersons, Peteris Stradins, Sandis Maurins, Juta Kroica

**Affiliations:** 1Department of Biology and Microbiology, Riga Stradins University, LV-1007 Riga, Latvia; 2Department of Internal Diseases, Riga Stradins University, LV-1007 Riga, Latvia; 3Center of Nephrology, Pauls Stradins Clinical University Hospital, LV-1002 Riga, Latvia; 4Department of Surgery, Riga Stradins University, LV-1007 Riga, Latvia; 5Centre of Cardiac Surgery, Pauls Stradins Clinical University Hospital, LV-1002 Riga, Latvia; 6Joint Laboratory, Pauls Stradins Clinical University Hospital, LV-1002 Riga, Latvia; 7Department of Infection Prevention and Control, Pauls Stradins Clinical University Hospital, LV-1002 Riga, Latvia; 8Center for Infectious Diseases, Queen Astrid Military Hospital, B-1120 Brussels, Belgium; 9Laboratory for Molecular and Cellular Technology, Queen Astrid Military Hospital, B-1120 Brussels, Belgium; 10Institute of Radiology, Pauls Stradins Clinical University Hospital, LV-1002 Riga, Latvia

**Keywords:** phage therapy, biofilm, LVAD infection, phage resistance, multidrug resistance, case report, PET/CT

## Abstract

There is considerable interest in the use of bacteriophages (phages) to treat *Pseudomonas aeruginosa* infections associated with left ventricular assist devices (LVADs). These infections are often challenging to manage due to high rates of multidrug resistance and biofilm formation, which could potentially be overcome with the use of phages. We report a case of a 54-year-old man with relapsing multidrug-resistant *P. aeruginosa* LVAD driveline infection, who was treated with a combination of two lytic antipseudomonal phages administered intravenously and locally. Treatment was combined with LVAD driveline repositioning and systemic antibiotic administration, resulting in a successful outcome with clinical cure and eradication of the targeted bacteria. However, laboratory in vitro models showed that phages alone could not eradicate biofilms but could prevent biofilm formation. Phage-resistant bacterial strains evolved in biofilm models and showed decreased susceptibility to the phages used. Further studies are needed to understand the complexity of phage resistance and the interaction of phages and antibiotics. Our results indicate that the combination of phages, antibiotics, and surgical intervention can have great potential in treating LVAD-associated infections. More than 21 months post-treatment, our patient remains cured of the infection.

## 1. Introduction

For patients suffering from advanced chronic heart failure, the left ventricular assist device (LVAD) is used as a bridge to heart transplantation or, in some cases, as destination therapy [1]. The annual incidence of LVAD implantation appears to be on the rise in recent years [2]. The main complications of LVAD are bleeding, thrombosis, pump failure, and infection, with the latter occurring in 50% of patients [3]. Like other implantable devices, infections associated with LVADs are complex to treat and often require device repositioning [4]. *Pseudomonas aeruginosa* is a frequently encountered pathogen that causes LVAD driveline-associated infections, which are challenging to manage due to the bacterium’s high rates of multidrug resistance and biofilm formation. These infections can lead to increased mortality rates, necessitating surgical intervention with debridement as a common treatment option [5].

Bacteriophages, or phages, are viruses that infect and kill bacteria by replicating and lysing the host cell. They offer a promising approach to treating *P. aeruginosa* biofilm-associated infections, as they can produce enzymes such as polysaccharide depolymerase that degrade the biofilm and thus destroy it [6]. Proper use of antibiotics and surgical intervention, in combination with phages, can lead to a higher treatment success rate. The lack of effective antibiotics has led to a more common use of phages, mainly in combination with conventional antimicrobials [7].

In the literature, we found 10 case reports using phages for LVAD-associated infections; the information is summarised in Appendix A. The main causative agents of these infections were as follows: in five cases, *S. aureus*; in one case, *S. aureus* and *P. mirabilis*; and in four cases, *P. aeruginosa*, of which three were MDR strains. Treatment in four cases involved surgery, antibiotics, and phage application [8,9,10,11]; in five cases, antibiotic and phage application [9,12,13,14]; and in one case, surgery and phage application [15]. These results show that phage therapy is commonly supplemented with antibiotics, and phages are rarely used as a single antimicrobial agent. Of the 10 patients, only 4 patients were cured and bacterial eradication was achieved. Some reasons for treatment failure were lack of lytic activity against the causative agent, as described by Püschel et al. [15], and neutralizing antibody development, as in a case of a 60-year-old male described by Aslam et al. [9]. The treatment modalities involved intravenous phage application and/or local application; the duration of treatment varied from 1 day to approximately 14 weeks. The phage concentration for intravenous application ranged from 10^7^ to 10^11^ PFU/mL, and for topical application, from 10^6^ to 10^9^ PFU/mL. The data in the literature show that the application of phage therapy in these cases is variable without consensus and clear evidence of the most suitable application for an effective outcome.

The presence of adverse events is another factor that contributes to the possible use of novel treatment applications. In case reports where phage therapy was used for LVAD infection treatment, mild nausea was noted in a case described by Mulzer et al. [8]. In a case of phage treatment for an 82-year-old man published by Aslam et al. [9], using high intravenous phage concentrations at a concentration of 10^11^ PFU/mL, the patient developed fever, wheezing, shortness of breath, and symptoms disappeared when the phage concentration was decreased to 10^10^ PFU/mL. In another case described by Tkhilaishvili et al. [10], increased liver markers were observed, but they returned to a normal range after lowering the dose of phages.

Although there are several case reports for the application of phage therapy in LVAD infection treatment, several gaps are still present, such as the dosage and routes of phage application, duration of treatment, combination of antibiotics, and development of phage resistance. Data suggest that the development of phage resistance in bacteria can cause trade-offs, potentially leading to reduced virulence and/or increased susceptibility to antibiotics [16]. Currently, there is a lack of clinical trial data on the use of phages for LVAD-associated infections. Therefore, the current understanding is based on individual clinical cases with varying outcomes.

In our article, we focus on the use of a combined treatment of phages, antibiotics, and surgical treatment for LVAD driveline infection. We present evidence of in vitro development of phage resistance that could interfere with infection eradication and makes it important to add antibiotics for treatment. The role of phages in biofilm eradication is not completely understood; therefore, we investigated the phage antibiofilm effect and also used surgical debridement for the patient’s treatment.

## 2. Case Description and Diagnostic Assessment

In November 2016, a 50-year-old male patient was admitted to our institution for LVAD HeartMate 3 (HM3) device implantation as a bridge to heart transplantation candidacy due to dilatation cardiomyopathy with severe end-stage heart failure (INTERMACS profile 1). In 2017, the patient was placed on the heart transplant waiting list.

In October 2020, the patient was readmitted with purulent discharge from the LVAD HM3 driveline exit site, inflammation of the exit site (Figure 1A), febrile temperature, and elevated inflammatory markers—C-reactive protein (CRP) 44 mg/L.

The symptoms mentioned first appeared 46 months after LVAD HM3 implantation. Immediate antibacterial therapy with piperacillin/tazobactam was started for two weeks. The exit site wound swabs were positive for *P. aeruginosa* (Table 1). After wound improvement, antibacterial treatment was changed to prolonged suppressive ciprofloxacin therapy, and the patient was discharged at the end of October 2020.

Nineteen weeks later, in March 2021, the patient was hospitalized again due to increased purulent discharge and fistula formation along the driveline (Figure 1B). The results of the wound swab showed the presence of multidrug-resistant (MDR) *P. aeruginosa* with no alternative oral antibiotics available. Intravenous antibiotic therapy with colistin was initiated with a loading dose of 9 million and then 3 million international units (IU) three times a day and continued until surgical intervention. A decision was made to prepare the patient for driveline repositioning. Fluorine-18-fluorodeoxyglucose positron emission tomography integrated with computed tomography (^18^F-FDG PET/CT) showed an active metabolic process along the driveline up to the level of the abdominal muscle, with a slight infiltration of the rectus abdominis muscle, representing infection.

To enhance the likelihood of a successful treatment outcome, phages were applied locally and intravenously during the intraoperative and postoperative phases. This decision was made based on a lack of response to previous treatments. Additionally, ceftazidime/avibactam and amikacin were the only remaining intravenous alternatives (Table 1). Two lytic phages, PNM and PT07, were shipped from the Queen Astrid Military Hospital (QAMH, Brussels, Belgium) to Riga (Latvia), and the phage treatment modality was discussed with the local treatment team and QAMH specialists. The treatment was conducted in accordance with Article 37 of the Declaration of Helsinki [17], and written informed consent was obtained from the patient prior to the procedure.

## 3. Therapeutic Intervention

On 16 June 2021, the operation began with extensive tissue debridement along the course of the driveline, including partial removal of the rectus abdominis muscle. During debridement, multiple wound swabs were cultured to rule out undetected microorganisms and to understand the depth of the infectious process roughly and retrospectively. To increase adhesion in subcutaneous tissue, the outer layer of the LVAD driveline is covered with velour, which complicates the chance of eradication of microorganisms. Therefore, the driveline’s outer layer was removed and sent for microbiological investigation (Figure 1D). The operation was continued with wound irrigation and local treatment with Prontosan^®^ solution (B. Braun, Germany) with betaine surfactant and 0.1% Polyaminipropyl Biguanide (Polihexanide) as active substances. Then, a new subcutaneous canal was prepared in the anterior abdominal wall to reposition the driveline. To reduce the chance that fluid or tissue material might enter the new modular cable connector and ensure sterility during repositioning, a sterile ultrasound probe cover was used to cover the sides of the subcutaneous tunnel. The new subcutaneous canal and the previous canal infected with *P. aeruginosa* were irrigated with 250 mL 0.9% NaCl and then with 250 mL 4.2% NaHCO_3_, to make the surrounding environment more alkaline. Five minutes later, 50 mL of phage suspension consisting of PNM and PT07, each at a concentration of 10^7^ plaqueforming units (PFU)/mL, was applied to each wound. A new modular cable already connected to the LVAD controller was guided through the subcutaneous tunnel. The driveline was temporarily disconnected from the old modular cable and connected to the new one. After ensuring that the hemodynamics of the patient were stable and the LVAD was running, the driveline was repositioned through the subcutaneous tunnel. An 8-Fr catheter was inserted along the driveline to administer the phage solution. The wound was left open for secondary healing (Figure 1E).

Intravenous application of phages PNM and PT07 that showed lytic activity against patient isolate (Table 1), with a titer of 10^7^ PFU/mL each, started 2 h before surgery using an infusion pump at a rate of 13 mL/h for 6 h through a central venous catheter with a total volume of 80 mL, which was repeated once per day for 8 days. On the next day after surgery, the wound was rinsed using an 8-Fr catheter with 50 mL of PNM and PT07, with a titer of 10^7^ PFU/mL each, and this procedure was continued daily for three days. Prior to local application of phages, the wound was rinsed through the catheter with 250 mL 0.9% NaCl, and then with 250 mL 4.2% NaHCO_3_. Intravenous antibiotic therapy consisting of ceftazidime/avibactam 2.5 g over two hour infusion three times a day and amikacin 750 mg (7.5 mg/kg weight) over 60 minute infusion two times a day was started 2 h before the operation. Amikacin was continued for four weeks, and ceftazidime/avibactam for six weeks (Figure 2). Intraoperative wound samples showed the presence of *P. aeruginosa* at all wound levels (Table 1). Changes in wound dressing and swabbing were performed daily and did not show the presence of *P. aeruginosa*. Six days after repositioning the driveline, the secondary healing wound did not reveal the presence of infection and was closed. During phage treatment, no adverse events were observed. The phage titer in the patient’s blood was stable for seven consecutive treatment days with a concentration of 10^2^ PFU/mL. Phages were no longer detected from the first day after cessation of phage administration (Figure 2).

## 4. Follow-Up and Outcomes

Six weeks after the driveline repositioning control, a ^18^F-FDG PET/CT scan was performed, showing slight residual metabolic activity in the most proximal part of the driveline, but to a lesser extent than before the driveline repositioning (Figure 3B). Considering that PET/CT was performed early after repositioning and most likely represented reactive changes, the wound showed no signs of inflammation, with inflammatory markers (CRP and white blood cell count) within normal limits. The patient was discharged on postoperative day 45.

Thirty-four weeks after the operation, another control PET/CT scan was performed, which showed no signs of significant metabolic activity (Figure 3C). The patient was regularly checked and showed no signs of recurrence 21 months after treatment (Figure 1C).

## 5. Materials and Methods

### 5.1. Bacterial Isolate Identification and Antimicrobial Susceptibility

*P. aeruginosa* isolated from the patient’s wound and LVAD exit-site discharges (Table 1), as well as a reference strain CN573, a host and propagation strain for PT07 and PNM phages, was used. CN573 is a bacterial strain isolated from bone marrow in the 1970s at the Eliava Institute of Bacteriophage, Microbiology and Virology. This strain has been characterised and used for the production of *P. aeurginosa* phages; it has the absence of temperate phages [18]. The Microflex LT system, a matrix-assisted laser desorption ionization-time-of-flight mass spectrometer (MALDI-TOF MS) manufactured by Bruker Daltonics GmbH & Co. KG in Bremen, Germany, was utilized to identify bacterial isolates using flex analysis software (version 3.4). The disk diffusion method was applied to determine the antimicrobial susceptibility of the *P. aeruginosa* isolates. The determination of minimum inhibitory concentrations (MICs) of antibiotics was performed using the MICRONAUT-S Pseudomonas MIC AST plates and the broth microdilution test (Merlin-Diagnostika, Bonn, Germany), or using an E-test (bioMérieux, France) for fosfomycin. The tests were performed, and the results were interpreted according to the European Committee on Antimicrobial Susceptibility Testing (EUCAST) breakpoints.

### 5.2. Phages, Phage Susceptibility, Efficiency of Plating (EOP)

Two lytic phages, podovirus PNM and myovirus PT07, were used for patient treatment. Phages were produced and provided by the Queen Astrid Military Hospital in Brussels, Belgium. Phage preparations were made in accordance with quality and safety requirements for small-scale phage productions for human treatments. These requirements involve genetic analysis to ensure the absence of unwanted genes, such as toxin-coding and antibiotic resistance genes, and the determination of the lytic nature of phages [18]. The quality control of the used phage stock was performed by Sciensano, the Belgian federal research institute for public health, to confirm phage identity and titer, pH, endotoxin level, and bioburden of the produced phage lots before the application of the phages. For patient treatment, both phages were shipped at a titer of 10^9^ PFU/mL, and before their clinical application, solutions were diluted to 10^7^ PFU/mL in 0.9% NaCl. For in vitro testing, PNM was provided at a concentration of 10^12^ PFU/mL and PT07 at 10^11^ PFU/mL.

The susceptibility of the target strain to the phages was determined using a spot test. One hundred microliters of an overnight bacterial culture was mixed with soft Tryptic Soy Agar (TSA; 0.7% agar) and poured on a TSA plate. After solidification of the top agar, 10 µL of the desired phage at a concentration of 10^7^ PFU/mL was spotted and left to dry for approximately 20–30 min. The plates were incubated for 16–18 h at 35 °C, and the result was visually evaluated as confluent lysis (++++), semiconfluent lysis (+++), partial lysis (++), individual plaques (+), or without lysis (-).

To determine the efficiency of plating (EOP), 100 µL of an overnight bacterial culture was mixed with 50 µL of the desired phage solution using serial phage dilutions from 10^−1^ to 10^−9^. Approximately 5–6 mL of previously melted 0.7% TSA was added and gently mixed, then poured on a TSA plate and grown for 16–18 h at 35 °C. Plaques were counted to determine the phage concentration in the bacterial isolate, and the titer was calculated. The EOP was calculated by dividing the phage titer of the tested isolate by the phage titer of its host *P. aeruginosa* strain (CN573). All experiments were performed in duplicate.

### 5.3. Biofilm Formation

Biofilm formation was studied using a crystal-violet assay. Briefly, a sterile 96-well flat-bottom microtiter plate (96-well TC plate; Suspension, F, Sarstedt, Germany) was used in the assay. The inoculum was obtained by making a suspension in Tryptic Soy Broth (TSB) of three to four colonies of TSA. The suspension was then cultured overnight and subsequently diluted to 1:100 in TSB and used for inoculum. Two hundred microliters of inoculum was added to each well, and the plate was incubated at 35 °C for 2, 4, 8, 16, 24, or 32 h. Only broth was used as a negative control. The wells were rinsed with 250 μL of sterile 0.9% NaCl after incubation. Then, the biofilm was stained with 200 μL of 0.1% crystal violet for 25 min, rinsed thrice with 250 μL of distilled water, and decolorized by the addition of 200 μL of 96% ethanol. Biofilm formation was measured spectrophotometrically (Tecan Infinite F50, Männedorf, Switzerland) at 600 nm (OD600) wavelength. Bacterial growth was considered to occur at an optical density value of >0.10. Each bacterial strain had more than 12 replications.

### 5.4. Detection of Bacterial Growth Suppression, Minimum Biofilm Eradication Concentration (MBEC), and Biofilm Prevention Concentration (BPC) Using Turbidity Reduction Assay

Bacterial strains were incubated at 35 °C overnight and mixed in TSB to reach a 1.0 MacFarland standard. For the inoculum, the broth was further diluted to 1:30 to achieve a concentration of 1.0 *×* 10^7^ colony-forming units (CFU)/mL. A sterile flat-bottom 96-well plate (Nunc™ MicroWell™ 96-Well, Nunclon Delta-Treated, Flat-Bottom Microplate, Thermo Fisher Scientific, Roskilde, Denmark) was filled with inoculum, 150 μL per well. Subsequently, the 96-well plate was covered with a 96-peg lid (Nunc™ Immuno TSP Lids) and incubated in a shaking table incubator (Infors^TM^ HT Ecotron, Basel, Switzerland) at 35 °C, 150 rpm for 24 h. The 96-peg lid was then transferred to a 96-well challenge plate containing 200 μL TSB per well of the desired phage or phage combination with phage titers of 10^7^–10^9^ PFU/mL. After 12 h of incubation at 35 °C, 150 rpm, suppression of bacterial growth was determined by measuring the optical density of the plate at 600 nm wavelength. The lid was transferred to another challenge plate with the same phages and their concentrations for 12 h of incubation to increase the phage effect. After a total of 24 h of incubation with phages, bacterial growth suppression was determined. The 96-peg lid was transferred to a fresh 96-well TSB plate, 200 μL per well, and sonicated for 25 min at 44 Hz using an ultrasonication bath (Model 08855-02, Cole-Parmer, Vernon Hills, IL, USA) to remove established biofilm. To recover the bacteria from the biofilm, the plate was covered with a sterile lid without pegs and incubated for 22 h at 35 °C stationary. To determine the minimum biofilm eradication concentration (MBEC), the optical density was measured at 600 nm wavelength.

The biofilm prevention concentration (BPC) was determined by inoculating bacteria and phages in their respective concentrations at the same time. Bacterial growth suppression was determined after 12 h of incubation of the inoculum plate. The plate was then changed to another challenge plate with phages and incubated for 12 h more, and after 24 h of incubation, again bacterial growth suppression was determined. The 96-peg lid was transferred to a 96-well sterile TSB plate and sonicated, then stationary incubated for 22 h at 35 °C, and the optical density for determining BPC was measured.

Between all steps after incubation, the 96-peg lid was rinsed in 0.9% NaCl, 200 μL per well, for 2 min and then transferred to the next 96-well plate. All 96-well plates used in the experiments were sealed with Parafilm^®^ (Bemis Company, Inc, Neenah, WI, USA). The tests were produced in at least 10 replicates. Sterile TSB was used for the negative control that has OD < 0.10; for the positive control, TSB with only bacteria was used.

### 5.5. Detection of Differences in Phage Susceptibility and Phage Resistance of PAP01 in Bacterial Growth Suppression, MBEC, and BPC Models

To test for changes in phage susceptibility, bacterial cultures from turbidity reduction assay in 96-well plates were recovered on Mueller Hinton Agar (MHA) with 5% sheep blood. This was performed once the microtiter plate incubation was finished, and OD was detected. Twenty-two bacterial cultures were taken randomly from different microtiter plates described in Section 5.4. using patient isolate PAP01. Cultures were taken from wells using a sterile 1 μL loop. Then the MHA inoculation was performed using the streaking method, and the plates were incubated for 24 h at 35 °C. Then three to four colonies from the MHA plate were inoculated in TSB and incubated overnight. Phage spot tests were performed using recovered bacteria as described in Section 5.2.

The cut-off value to determine phage resistance was calculated using a positive control (bacteria only). Bacteria were considered resistant to the phage if the measured OD > OD mean—(3 × SD) of the positive control. Phage resistance was detected after 12, 24, and 46 h with respect to the microtiter plate from which bacterial cultures were recovered. To show the difference between the measured mean optical density of the well (OD_well_) compared with the optical density of resistance cut-off value (OD_CF_), a ratio (OD_ratio_) was used OD_ratio_ = OD_well_/OD_R_. If the value was less than 1, bacteria were not considered resistant, but if the value was greater than or equal to 1, bacteria were considered resistant to phage.

### 5.6. Phage Concentration Detection in Peripheral Blood Samples

A phage plaque assay was performed to determine the titer of infective phages in peripheral blood samples that were collected 2 h after intravenous phage administration in plastic EDTA blood-drawing tubes. To execute the aforementioned assay, 100 µL of PAP01 suspension was mixed with 50 µL of the peripheral blood sample. The mixture was then introduced into 0.7% TSA soft agar to spread onto 1.5% TSA solid agar plates. The blood phage level, i.e., the detected concentration of phages, was expressed as PFU/mL. Phage was measured in blood, in duplicate, for 11 consecutive days. The detection range for this experiment was 10^1^–10^8^ PFU/mL.

### 5.7. ^18^F-FDG PET/CT Imaging

Whole-body ^18^F-FDG PET/CT scans were performed from the top of the head, including the extremities. Additionally, low-dose CT was performed to process attenuation correction (CT-AC) and used for anatomical co-registration of PET findings. ^18^F-FDG PET/CT images were acquired 60–75 min after the intravenous injection of ^18^F-FDG (3.3 MBq per kg), using an ^18^F-FDG PET/CT combined with a third-generation multi-slice spiral CT scanner with a dedicated full-ring PET scanner, which had a high-count-rate-capability lutetium-yttrium oxyorthosilicate (LYSO)-based camera with time-of-flight (TOF) technology. Emission scans of ^18^F-FDG PET/CT were acquired 45 s–2 min per bed in 3D mode. PET and CT images (non-corrected and attenuation-corrected) were evaluated in a rotating maximum-intensity projection and a cross-sectional plane view (transverse–sagittal–coronal). Images with increased focal uptake, with higher intensity than surrounding tissues, which did not correspond to the physiological distribution of ^18^FDG, were defined as positive.

### 5.8. Data Analysis

Statistical and graphical analyses were performed using GraphPad Prism software (version 9) and MS Excel (version 10). The results were analysed using the Mann–Whitney *U* test.

## 6. Results

### 6.1. Bacterial Isolates, Their Susceptibility to Antibiotics and Phages

In total, five bacterial isolates were used, all identified as *P. aeruginosa*. For further biofilm modeling and phage testing, the PAP01 isolate obtained from the driveline velour removed during surgery was used. For antimicrobial susceptibility results, phage susceptibility testing and EOP results, see Table 1.

### 6.2. Phage Effect against Planktonic Cells, Biofilm Eradication, Prevention of Biofilm Formation, and Bacterial Resistance to Phages

Patient isolate PAP01 showed weak biofilm production in crystal-violet assay (Appendix A). The results of the biofilm eradication test in PAP01 showed that only phage PNM alone at a concentration 10^9^ PFU/mL could decrease biofilm formation to some extent (Figure 4). Other phage concentrations of PNM and PT07, the combination of PT07 and PNM at any concentration, did not have a biofilm eradication effect. Planktonic cell growth after 12 h decreased for all phages and concentrations tested; however, after 24 h, this effect persisted only for PNM at concentrations of 10^7^–10^8^ PFU/mL and for a combination of phages at the tested concentrations. In all cases, except for PT07 concentrations 10^7^ versus 10^8^ PFU/mL, the bacterial growth suppression effect after 12 h was better when higher phage concentrations were used. This was not observed 24 h later, and was even the opposite was observed for PNM at concentrations 10^7^ versus 10^9^, and 10^8^ versus 10^9^ PFU/mL. Phage resistance developed in all cases after 24 h; however, it was less common in phage combination and reached 100% in bacterial cells recovered from biofilms for PT07 at all concentrations tested and for PNM at concentration of 10^7^ PFU/mL. In comparison, the phage biofilm eradication effect was achieved in the host strain CN573: the emergence of resistance was low, and it developed only against the PNM phage and was 20% (Appendix A).

The biofilm prevention capacity in PAP01 for PT07 and the combination of PNM with PT07 were determined at all tested concentrations. PNM alone did not prevent biofilm formation; on the contrary, the use of 10^7^ and 10^8^ PFU/mL concentrations led to a higher bacterial growth compared with untreated PAP01. Bacterial growth suppression was observed for all phages and their tested concentrations 12 and 24 h after incubation. Phage resistance developed for all phages tested in bacteria recovered from biofilms; it reached 100% for PT07 at all tested concentrations. Bacterial resistance was observed only after 46 h of incubation for PNM phage and a combination of both phages. The resistance rate to PNM and phage combination differed in cells recovered from the biofilm (Figure 5).

### 6.3. Differences in Phage Susceptibility in Bacteria Recovered from the 96-Well Plates Used in Turbidity Reduction Assay

We evaluated the emergence of changes in phage susceptibility in bacterial cultures recovered from 22 wells from different microtiter plates used in MBEC and BPC detection models for PAP01 isolate. In 1 case, bacteria could not be recovered. In 14 of the remaining 21 cases, the phage lytic effect decreased, and in 7 of 21 cases, “no change” in the lytic effect was observed. In the biofilm eradication model, “no change” in phage lytic activity was observed more frequently than in the biofilm prevention model (see Table 2).

## 7. Discussion

We present a case of persistent LVAD driveline infection with MDR *P. aeruginosa* that was successfully treated with local and systemic application of phages in combination with antibiotics and surgical treatment, including LVAD driveline relocation. No adverse events were observed. The application of phages in in vitro models showed that phage–bacteria interactions vary in planktonic cells versus bacterial biofilms. In an in vitro model, phage therapy could not eradicate the biofilm produced by the patient’s bacterial strain, but it could prevent the formation of the biofilm. Therefore, proper surgical intervention with complete bacterial debridement is crucial. In complex biofilm-associated and MDR infections, proper investigation, including hybrid imaging such as ^18^F-FDG PET/CT, is helpful to understand the extent of the infection.

LVAD implantation has become more common in recent years, mainly as a bridge to heart transplantation. In addition to right ventricular failure, bleeding, thromboembolism, and pump malfunction, infection is one of the common complications observed [19]. Only 58.9% of patients with LVAD are estimated to not develop the first major infection in the first year after device implantation; at three years after device implantation, only 38.2% of patients have not had a major infection [2]. Driveline infections are among the most common infections associated with LVAD, with a prevalence of 18.8–100% of all infections related to LVAD [20]. The primary pathogens causing such infections are Gram-positive cocci. However, Gram-negative bacilli are becoming a common and concerning problem for patients with LVAD because they are frequently having multidrug resistance. In the ASSIST-ICD study in which 19 centers were involved, *P. aeruginosa* infections were detected in 13.7% of cases [21,22]. The standard treatment for driveline-associated infections involves systemic antibiotics; however, commonly, such an approach leads to treatment failure, and surgical relocation of LVAD driveline is necessary. In a study in Warsaw, Poland, the primary success rate with antibiotics was only 27%, driveline repositioning was needed for 73.1% of patients, and the mortality in these patients was 11.5% [4].

Multidrug resistance is a common feature in *P. aeruginosa* infections, ranging from 11.5 to 24.7%. In relapsing and chronic infections, treatment is challenging due to the lack of effective antibiotics and changes in the resistance pattern of *P. aeruginosa* during treatment [23]. In recent years, the application of phages has become increasingly attractive for the treatment of MDR and extensively drug-resistant (XDR) *P. aeruginosa* infections due to their ability to destroy extracellular matrix and to reach inner structures and cells in biofilms, and to increase the effectiveness of antibiotics. Phage therapy in combination with antibiotics has been used in several case reports for different indications involving *P. aeruginosa* infections [6].

Our results show that the topical and intravenous application of phages combined with antibiotics and surgical treatment can be most appropriate for the successful outcome of infections associated with LVAD. There are two more cases of MDR *P. aeruginosa* LVAD-associated infection treatment described using phages. In one case, the same treatment modality was used with clinical cure and bacterial eradication [10]. In the other case, only phages and antibiotics were used, and the treatment failed, with relapse of infection and development of phage resistance during treatment [14]. Our case also shows that intravenously applied phages in concentration 10^7^ were safe and did not elicit side effects similar to the results presented by Aslam et al. [9]. Therefore, for future phage applications, a dosage of 10^7^ PFU/mL for antipseudomonal phages could be optimal. However, only in 7 out of 10 published cases such information is given, which makes it hard to compare (Appendix A).

The application of phages in an in vitro biofilm model yielded varying results. Both phages, when used in the host (maternal) strain CN573 could lyse planktonic cells and eradicate established biofilm (Appendix A). In contrast, for the patient isolate, PAP01, biofilm eradication was not achieved (Figure 4). One of the obstacles that could explain these results is the bacterial susceptibility to the phages used, which was better for CN573 than for PAP01; for instance, the EOP for PAP01 using PNM was only 0.0005. When pre-adapted to host bacteria, as was the case for PNM and PT07 in CN573, phages are more efficient. Therefore, the best phage therapy results are expected to be achieved using personalized phage preparations, containing phages that were selected, or even pre-adapted, to better target the patient’s infecting strain(s) [24]. However, such an approach is time- and cost-consuming, making phage therapy harder to apply in a clinical setting. However, it is important to note that last-resort antibiotic treatment is also highly expensive and not always available. Another critical factor is the structure of the biofilm; according to our results, *P. aeruginosa* strain CN573 produced greater biofilm than strain PAP01, assuming that it would be easier for phages to eradicate the biofilm of PAP01. However, this was not observed and could be explained by differences in biofilm density. Hu et al. showed that phage penetration depends on biofilm density [25]. This is one of the limitations of our study because we did not determine the biofilm density, and other methods, such as confocal laser scanning microscopy, should be used to identify the biofilm density. This could clarify whether the weak biofilm eradication effect of the phages in PAP01 was associated with the density of the biofilm.

The development of phage-resistant bacterial strains can occur quickly both in vitro and in vivo [26]; by reducing the density of the biofilm, the presence of phage-resistant strains can be detected [6]. In our study, resistance to phages was observed in the biofilm eradication model, with an incidence rate ranging from 70% to 100% for PNM and PT07, as well as for their combinations, using three different concentrations. Therefore, this could explain their failure to eradicate the biofilm. Resistance was present in the biofilm prevention model but it was less common; for PNM, it was 80–100%; for PT07, 10–40%, but when using a combination of these phages, it was 10–20% (Figure 5). Strategies involving combined treatment of phages and antibiotics can lead to a better outcome because the development of resistance against one agent can elicit increased susceptibility to another agent. A study by Burmeister et al. showed that there is even a trade-off between phage resistance and antibiotic resistance, which means that in phage-resistant strains, a possible susceptibility to antibiotics can evolve [27]. However, the interaction of phages and antibiotics is complex and does not always exhibit a synergistic effect; on the contrary, even antagonistic effects could be observed. The effect of phage-antibiotic combinations depends on the administration order, the concentration of phage, and the antibiotic’s and the phage’s mechanism of action [28,29]. A crucial limitation of our study is the lack of investigation of antibiotics used in different combinations with phages and their antibiofilm effect.

The current standard treatment of LVAD biofilm-associated infections involves debridement and antibiotic therapy. Surgical intervention is crucial to mechanically remove and eliminate biofilm from the driveline [20]. Similarly, in our case, debridement and repositioning of the driveline were performed and, most likely, were the cornerstone for biofilm eradication; however, the formation of new biofilm and the development of bacteremia or septicaemia from the residual bacterial cells were prevented with antibiotics and phages.

Another reason to supplement antibiotic treatment with phages is the rapid development of antibiotic resistance in *P. aeruginosa.* This happens due to the presence of intrinsic and acquired resistance mechanisms [30]. Bacterial isolates of our patient also showed changes in the pattern of resistance to antibiotics throughout the time of infection (Table 1). Therefore, we cannot be sure that resistance to amikacin and ceftazidime-avibactam will not occur during treatment. This explains the need to use other effective agents as phages in MDR bacterial infections.

## 8. Conclusions

We describe a successful case of phage treatment of MDR *P. aeruginosa* LVAD driveline infection that was performed together with surgical intervention, debridement, and antibiotic therapy. More research should be conducted, focusing on phage dosage, duration of treatment, phage interaction with antibiotics, and the impact of phage resistance.

## Figures and Tables

**Figure 1 viruses-15-01210-f001:**
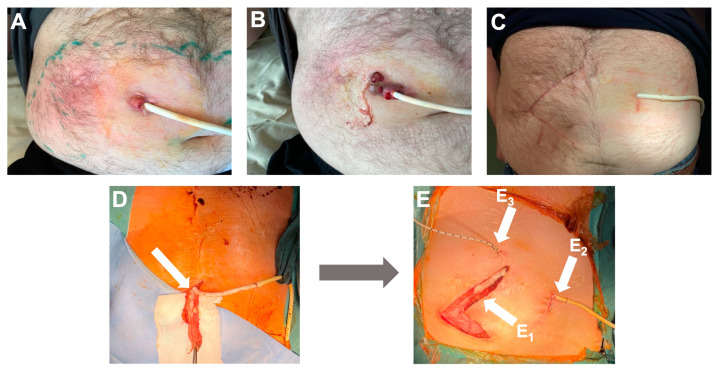
The appearance of the wound before, during, and after surgery. (**A**) The wound in October 2020 before antibacterial treatment. (**B**) The wound in March 2021 before antibacterial treatment. (**C**) The wound in October 2021 with no local signs of infection, four months after phage treatment and surgery. (**D**) Removal of the velour from the LVAD driveline during surgery; velour indicated with an arrow. (**E**) In a picture of surgery, arrow E_1_ is pointing at the previous driveline canal, arrow E_2_ indicates the exit site of the new driveline canal, and arrow E_3_ indicates an 8-Fr catheter for local phage infusion in the new canal.

**Figure 2 viruses-15-01210-f002:**
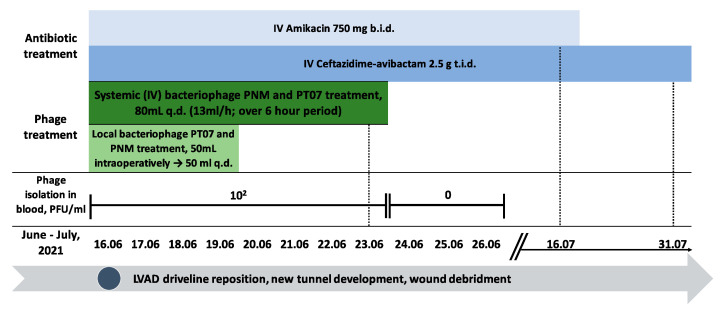
Timeline showing antibiotic and phage treatment modalities, surgical interventions, and phage concentrations detected in the patient’s blood. The blood samples for phage concentration detection were taken 2 h after intravenous phage administration.

**Figure 3 viruses-15-01210-f003:**
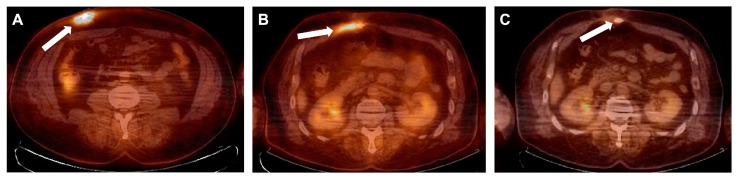
^18^F-FDG FDG PET/CT images. (**A**) April 2021—condition after implantation of the LVAD. Hypermetabolic changes in the skin, subcutaneous tissue, and muscles along the way of the LVAD cable in the anterior abdominal wall from its entrance gate to the rectus abdominis muscles corresponding to (most likely chronic) inflammation. No other pathological hypermetabolic changes during the whole-body exam or in other implanted devices or cables, and no any dissemination focuses. (**B**) July 2021—comparison with the previous PET/CT exam; hypermetabolic changes along the way of the LVAD cable are seen to a much less extent but visible near the rectus abdominis muscles and in subcutaneous tissue, most likely representing reactive changes due to recent cable repositioning. (**C**) February 2022—slight hypermetabolic changes along the way of the LVAD cable are seen with less metabolic activity and extent, but likely without clinical significance. No other dissemination focuses of infection are seen during the whole-body examination. Hypermetabolic changes are indicated with arrows.

**Figure 4 viruses-15-01210-f004:**
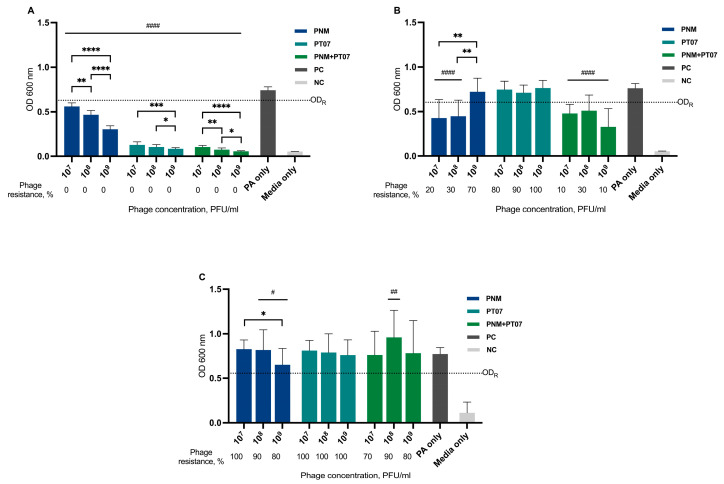
Bacterial growth suppression and MBEC of PNM, PT07, or both combined against PAP01 in a biofilm eradication model. The bars represent mean values + standard deviations. The number of phage-resistant strains in % is represented below the bars. (**A**) Bacterial growth suppression after 12 h; (**B**) Bacterial growth suppression after 24 h; (**C**) MBEC; PC—positive control, untreated PAP01; NC—negative control, media only; OD_R_ line represents the cut-off value to determine phage resistance; bars with a hash sign are statistically different from the positive control, and bars with an asterisk represent the statistical difference between concentrations of the same phage; */# *p*-value < 0.05, **/## *p*-value < 0.01, *** *p*-value < 0.001 and ****/#### *p*-value < 0.0001.

**Figure 5 viruses-15-01210-f005:**
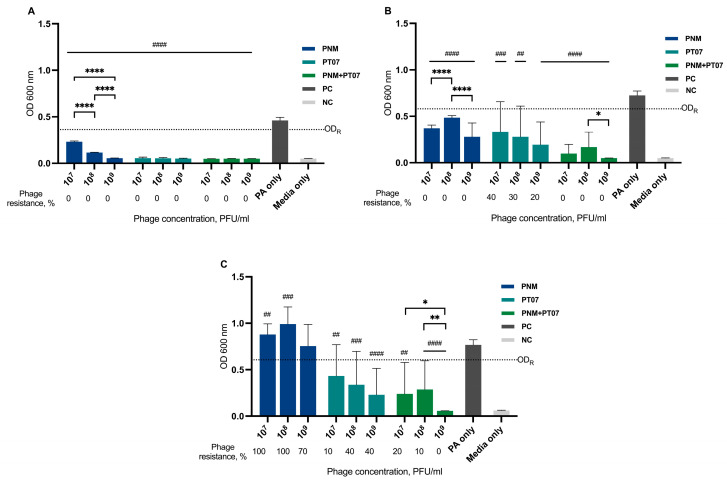
Bacterial growth suppression and BPC of PNM, PT07, or both combined against PAP01 in a biofilm prevention model. The bars represent mean values + standard deviations. The number of phage-resistant strains in % is represented below the bars. (**A**) Bacterial growth suppression after 12 h; (**B**) Bacterial growth suppression after 24 h; (**C**) BPC; PC—positive control, untreated PAP01; NC—negative control, media only; OD_R_ line represents the cut-off value to determine phage resistance; bars with a hash sign are statistically different from the positive control, and bars with an asterisk represent the statistical difference between concentrations of the same phage; * *p*-value <  0.05, **/## *p*-value < 0.01, ### *p*-value <  0.001 and ****/#### *p*-value < 0.0001.

**Table 1 viruses-15-01210-t001:** Antibiotic and phage susceptibility of the patient’s consecutive *P. aeruginosa* isolates.

Strain	CN573	PA01	PA02	PA03	PAP01	PAP02
Type of the isolate	Phage host strain	Discharge from LVAD driveline exit site	Discharge from LVAD driveline exit site	Discharge from LVAD driveline exit site	Velour from the driveline	Velour from the driveline
Isolation time		4 October 2020	5 March 2021	19 Apri 2021	16 June 2021	16 June 2021
**Antibiotics**						
AMK	S (MIC ≤ 4)	S	S	S	S (MIC ≤ 4)	S (MIC ≤ 4)
FEP	I (MIC = 4)	ND	ND	ND	I (MIC = 2)	I (MIC = 4)
CAZ	I (MIC = 2)	I	R	I	I (MIC = 2)	I (MIC = 2)
CAZ/AVI	S (MIC ≤ 1)	ND	ND	ND	S (MIC ≤ 1)	S (MIC ≤ 1)
CIP	I (MIC = 0.25)	I	R	R	I (MIC = 0.25)	I (MIC = 0.25)
CST	S (MIC = 2)	ND	S	S	S (MIC ≤ 1)	S (MIC = 2)
FOF	R (MIC > 128)	R (MIC > 128)	R (MIC > 128)	ND	R (MIC > 128)	R (MIC > 128)
IPM	I (MIC ≤ 1)	I	R	R	I (MIC ≤ 1)	I (MIC ≤ 1)
MEM	S (MIC = 1)	S	R	I	S (MIC ≤ 0.125)	S (MIC = 1)
TOB	S (MIC = 0.5)	ND	R	R	S (MIC = 0.5)	S (MIC = 0.5)
TZP	I (MIC = 8)	I	R	I	I (MIC = 4)	I (MIC = 8)
Phage PNM	++++	ND	ND	++ EOP = 0.001	++ EOP = 0.0005	ND
Phage PT07	++++	ND	ND	+++ EOP = 0.1	+++ EOP = 0.1	ND

ND, not determined; R, resistant; S, susceptible; I, susceptible, increased exposure; MIC, minimum inhibitory concentration, mg/l; AMK, amikacin; FEP, cefepime; CAZ, ceftazidime; CAZ/AVI, ceftazidime/avibactam; CIP, ciprofloxacin; CST, colistin; FOF, fosfomycin; IPM, imipenem; MEM, meropenem; TOB, tobramycin; TZP, piperacillin/tazobactam; ++++, confluent lysis; +++, semiconfluent lysis; ++, partial lysis; EOP, the efficiency of plating.

**Table 2 viruses-15-01210-t002:** Differences in phage susceptibility from recovered bacteria in PAP01 models.

		Resistance Detection Time, h	Phage Tested in Well, PFU/mL	OD_ratio_	PNM Susceptibility	PT07 Susceptibility	Susceptibility Change
Biofilm Eradication Model	1	12	PT07, 10^7^	0.32	ND	+++	No change
	2	12	PT07, 10^8^	0.25	ND	++	Decrease
	3	12	PNM, 10^7^	0.94	+	ND	Decrease
	4	24	PT07, 10^7^	1.53	ND	+++	No change
	5	24	PNM, 10^9^	1.54	++	ND	No change
	6	24	PT07 + PNM, 10^8^	1.12	-	++	Decrease
	7	24	PT07 + PNM, 10^9^	0.10	-	+++	Decrease
	8	46	PT07, 10^7^	1.54	ND	+++	No change
	9	46	PNM, 10^8^	1.85	++	ND	No change
	10	46	PT07 + PNM, 10^8^	2.32	-	-	Decrease
Biofilm Prevention Model	11	24	PT07, 10^7^	1.31	ND	++	Decrease
	12	24	PT07, 10^8^	1.41	ND	++	Decrease
	13	24	PT07, 10^8^	0.09	No bacteria recovered		
	14	24	PNM, 10^8^	0.84	-	ND	Decrease
	15	24	PT07 + PNM, 10^7^	0.45	-	++	Decrease
	16	24	PT07 + PNM, 10^8^	0.71	-	-	Decrease
	17	46	PT07, 10^7^	1.29	ND	+++	No change
	18	46	PT07, 10^7^	0.77	ND	++	Decrease
	19	46	PT07, 10^8^	1.24	ND	++	Decrease
	20	46	PNM, 10^9^	0.72	-	ND	Decrease
	21	46	PT07 + PNM, 10^7^	0.33	+	+++	Decrease
	22	46	PT07 + PNM, 10^8^	0.33	++	+++	No change

ND, not determined; Resistance detection time is the time after incubation of the 96-well plate from which the bacterial cultures were recovered; OD*_ratio_*, is the ratio between OD_well_ to OD_CF_; OD_well_, measured OD of the well at 600 nm; OD_R_, calculated OD cut-off value for resistance detection of the positive control; +++, semiconfluent lysis; ++, partial lysis; +, individual plaques; -, without lysis.

## Data Availability

The data presented in this study are available on request from the corresponding author. The data are not publicly available due to patient privacy.

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
