# Peer review of "Successful Bacteriophage-Antibiotic Combination Therapy against Multidrug-Resistant Pseudomonas aeruginosa Left Ventricular Assist Device Driveline Infection"

_viruses, 2023, doi:10.3390/v15051210_

Round 1

Reviewer 1 Report

Since the results are based on a case report, the title should include that information as it's currently misleading.

In the Abstract, there seems to be confusing contradictory information where the combination treatment of the patient is said to eradicate the target bacteria, but the lab experiments showed that eradication of biofilm was not possible – it would be helpful to differentiate the treatments by stating that the ‘phage only’ treatment of biofilms in vitro didn’t result on biofilm eradication.

The authors should consider presenting their paper in two halves: first outlining how the case report was conducted from identifying the patient with the issue to treating him. They then could describe the in-vitro experiments used to characterize the clinical isolates and their interactions with the phages used. My biggest critique of the paper as it’s currently presented is that the in-vitro experiments are not appropriate representations of the clinical case since the experiments only test ‘phage alone’ treatments and don’t show/compare them with any antibiotic and phage treatments.

For the Introduction section, the authors should also consider focusing on what their paper will be presenting and supporting, rather than making statements, such as “Although phage therapy has been used for a long time to treat bacterial infections, there is still limited knowledge about optimal application routes and dosages, the possible development of resistance to phages, and the appropriate use of phages in combination with antibiotics.” The paper doesn’t provide any evidence or knowledge to fill the gap in such knowledge so it seems to be a moot statement to make.

Other points to consider when revising the paper:

The background provided, although includes pertinent information on the need for LVADs and the current issues related to their use, the last paragraph is misleading in that it suggests that clinical trials would be able to fill in the gap in knowledge for the optimal use of phage applications – this seems highly unlikely due to the number of variables needed to be tested to fill in the gaps e.g., each phage or phage combinations would need to be tested for their application routes, dosages, development of resistance, combination with antibiotics used in standard care etc. It would be more appropriate for the authors to focus on how the results from their experiments can help towards improving the use of LVADs.

Lines 84-85 states that “no effective antibiotic options were available, with ceftazidime/avibactam and amikacin being the only remaining intravenous alternatives”. This needs to be clarified as to why would the antibiotics mentioned be used if there were no effective antibiotic options available. It’s also unclear if the 2 phages used on the patient were tested against the MDR causing the infection. This wasn’t clearly outlined in the paper.

In the Methods section, under the Therapeutic intervention (e.g., line 133) although the amount of antibiotics administered is stated, there is no information about the dose of antibiotics used, which would be helpful values for comparison to other studies.

Lines 151-152, the authors mention ‘inflammatory markers lysing within normal limits’ but they don’t provide a list of examples of which makers they are referring to… this information is not very helpful in its current presentation.

Lines 155-156, “The patient is regularly checked and shows no signs of recurrence months after treatment”, this should be in the past tense.

Line 188, change ‘Phages’ to “Phage preparations’

Line 190’, change ‘excluding’ to 'absence of' since the former implies that the genes were removed.

Line 199, change ‘susceptibility of the phages’ to ‘susceptibility of the target strain to the phages’.

Very little information is provided about the phage host strain CN573, especially when compared to PA01 in Figure 4. More information needs to be provided in the methods section to help evaluate/compare the differences between strains as a lack of information can result in biased assumptions.

Figures 4, 5, and 6 need graphs that have the same range for the y-axis (within each figure) to make it easier to compare the results between the graphs. The current setup seems misleading. Also, the 'Media only' bar is over '0' value, since that wasn’t used as the 'tare' agent, it would be helpful to know, somewhere in the methods section, what was used to ‘zero’ the OD value.

Table 2 is confusing, as it uses ‘N/A’ as a key. What is the difference between 'N/A' not available, and Not Determined? Why is the result N/A? This makes it a little confusing to understand the data presented. Was the data not available because it wasn’t taken (i.e., Not Determined) or the data is missing and not available?

In the Discussion section, for lines 402-403, provide the evidence identified from the results in this paper that supports the need to use hybrid imaging such as FFDG PET/CT. The authors suggest that the imaging is able to determine if an infection has been eradicated, which in my experience using other imaging systems (i.e., the IVIS) seems highly improbable.

Lines 473-475, seems to assume that phage are able to degrade biofilms when that is dependent on the production of the phage encoding for degrading enzymes.

Overall, the research design for the in-vitro experiments are not adequate comparisons to the clinical treatment, the results were not clearly presented and could be improved by, and the discussion could be used to explain/justify the logic of needing to use phage therapy, the specific phages used, and the route of their administration. It currently falls short on providing helpful information to the phage therapy community.

Author Response

Dear reviewer,

Thank you for your review, please see for our answers and corrections word doucment with point by point answers.

Reviewer 2 Report

This article describes a case report for the treatment of a patient with a LVAD infection caused by P. aeruginosa and excludes a good in vitro characterization of phage sensitivity. I am always supportive of publishing clinical case reports such as this and the authors did a nice in vitro characterization of the clinical isolates; however, the authors are trying to fit a case report, extensive in vitro assays, and a review of the literature into one publication, which dilutes its focus and makes it very long. I recommend that this paper be published with revisions, mostly for methodological explanations and terminology, and for minor for minor English corrections and length, as it is quite long (reduction of results to only clinical isolates). Several methodological points to be reworded and clarified prior to accepting for publication. Please see comments both below and in the accompanying .pdf.

Major – MIC: (Methods & Figures 5/6/7, Table 2, throughout). The term “MIC” in not appropriate for the experiments performed by the authors in relation to phage, both in the experimental design (ie. growth on PEGs) and concept. Given the use of MIC as established measure for antibiotic sensitivity testing (AST) and the different methods/concepts used here, it is not analogous for phage and could be misleading by mixing terminology. Bacterial growth curves in relation to phage, which is essentially what the authors are showing by measuring the OD at different timepoints, is typically described as suppression of bacterial growth. I ask the authors address this by changing the term to something that is not MIC or simply describing what it is – a turbidity reduction assay. (Additional refutes to the MIC concept, a timepoint of 12 hours is not used for MIC in AST and the growth on PEGs is also specific to the authors experiments.)

In general, the authors cannot constitute bacterial lysis from optical density, as it is a proxy measure for bacterial growth (or absence of growth) – to determine if bacteria were lysed, viable counts would have to be performed. Please change the wording to reflect what the experiments measured – a significant difference in turbidity.

Length/Structure: Overall the article is quite long and authors should make a general effort to be more concise or focus on information important for the case report (eg. And not for reference strains). Section 6.2 and Figure 4 and 5 should be relegated to supplementary material.

Lines 422-454/Table 3: This is introductory material, as it is new information and not directly discussed in relation to the authors’ case and work. Please move to the introduction and shorten.

Lines 501-510: Please remove, this is not the subject of the manuscript.

Lines 457-500: This is a poorly structured paragraph with too many different pieces of information. Please decide, in function of other comments, what information must stay or be removed, and break different concepts into separate paragraphs.

Methods:

The authors description of inoculating wells and overnight cultures in respect to number of colonies or what is diluted is often confusing. Eg. Lines 215-217: “inoculated 3-4 colonies from an overnight culture.. and diluted”. Does this mean 3-4 colonies were used to start the ON culture? Or 3-4 colonies from an agar plate, grown ON, were inoculated per well?

Also at Lines 260-265 and elsewhere. Lines 395-402 where authors discuss the difference in a lab strain versus patient strain is another example of what should be removed to reduce the length of the article .

Figures/Tables:

Please ensure the resolution of the figures is sufficient, as some legends appeared blurry.

Add cutoff line for phage resistance to graphs

Figure 2 indicate time points at which blood was taken to measure phage titer

Why is there a Table in the Discussion? This should either be part of the Introduction and possibly put as supplementary material to reduce the length.

Author Response

Dear reviewer,

Please see the changes in the new manucsript and in the word document with point by point answers.

Round 2

Reviewer 1 Report

The changes made to the current draft have improved the readability of the paper and the information provided as a case report could be helpful to those interested in using phage therapy; however, my previous critique referring to a lack of connection between the in-vitro work (i.e., only phage treatments were tested) and the case report (i.e., phage and antibiotic treatment used) still remains.

Author Response

Dear reviewer,

I have made the changes according to your comments and have written the answers to your other questions. Please see the document in attachment.

Reviewer 2 Report

Dear Authors,

Thank you for your detailed revision and modifications. I am happy to suggest your paper for publication. 

Author Response

Dear reviewer,

Thank you, I have made slight changes according to other reviewer comments for antibiotic dosage calculations and infusion rates for patient treatment. See line 334-336.
